# The Geometry of Multilingual Language Models: An Equality Lens

**Cheril Shah, Yashashree Chandak**
PICT
{shahcheril311,chandakyashashree304}@gmail.com

**Manan Suri**
NSUT
manansuri27@gmail.com

## Abstract

Understanding the representations of different languages in multilingual language models is essential for comprehending their cross-lingual properties, predicting their performance on downstream tasks, and identifying any biases across languages. In our study, we analyze the geometry of three multilingual language models in Euclidean space and find that all languages are represented by unique geometries. Using a geometric separability index we find that although languages tend to be closer according to their linguistic family, they are almost separable with languages from other families. We also introduce a Cross-Lingual Similarity Index to measure the distance of languages with each other in the semantic space. Our findings indicate that the low-resource languages are not represented as good as high resource languages in any of the models

## 1 Methodology

We use the XNLI-15way dataset Conneau et al. (2018) and sample 300 parallel sentences across the 15 languages for our analysis. We use common multilingual transformers, mBERT Devlin et al. (2019), MiniLM Wang et al. (2020), and XLMR Conneau et al. (2020a). More details about the models and dataset are available in Appendix A. We study the geometric properties of multilingual models using three methods:

1) We visualise the embedding space of a group of languages by taking the top 3 PCA components.

2) **Cross-lingual Similarity Index** $\Gamma$**:** There have been many approaches to compute cross-lingual similarity such as Liu et al. (2020), however due to the extremely high anisotropy(average cosine similarity of any two randomly sampled words in the dataset) Ethayarajh (2019) of models like XLMR, it is difficult to make conclusions using cosine similarity. Thus we introduce a metric to quantify cross-lingual similarity. For a pair of languages, $l1, l2$, we take embeddings $s_{(l1,i)}, s_{(l2,i)} \forall i \in [0,299]$ using model $\mathcal{M}$ and calculate the Cross-lingual Similarity Index as follows:

$$\Gamma_{l1,l2} = \frac{\frac{1}{n}\sum_{i=1}^{n} cosine(s_{(l1,i)}, s_{(l2,i)})}{Anisotropy(\mathcal{M})} \qquad (1)$$

For a model $\mathcal{M}$, $\Gamma$ ranges from $-1/Anisotropy(\mathcal{M})$ to $1/Anisotropy(\mathcal{M})$. Low model anisotropy and high average cosine similarity is ideal, therefore higher $\Gamma$ is ideal. Positive and negative values of $\Gamma$ correspond to average directional orientation between embeddings. $|\Gamma| \leq 1/Anisotropy(M)$ would mean that the average similarity is less than the similarity for random words.

3) **Language Separability** $\Phi$**:** We study the separation of different languages in embedding space by treating all the points belonging to a language as a single cluster and calculating the pairwise Geometric Separability Index Thornton (2008) between two languages.

## 2 RESULTS

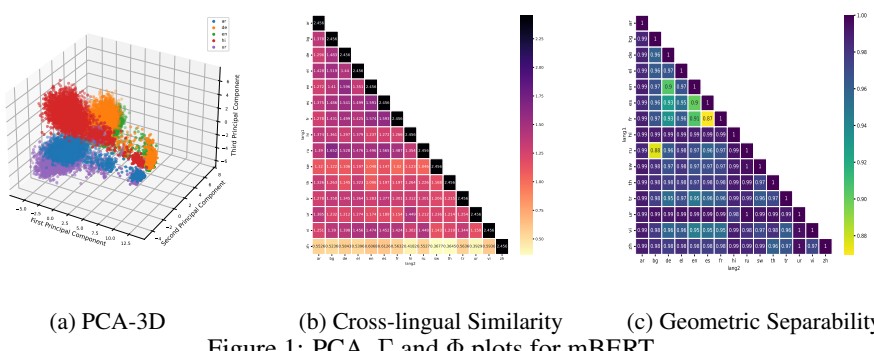

(a) PCA-3D  (b) Cross-lingual Similarity  (c) Geometric Separability

Figure 1: PCA, $\Gamma$ and $\Phi$ plots for mBERT

**PCA Analysis**: Figure 1a depicts the plots of one group (Hindi, Urdu, German, English, Arabic) for each model, while the remaining plots are available in the AppendixA.4. mBERT's word vectors demonstrate significant isolation, with the languages occupying different orientations in space. Conversely, MiniLM and XLMR's word embeddings appear less spread out, owing to their high anisotropy, but the languages still occupy different affines relative to one another. It is interesting to note that in XLMR, low-resource languages such as Urdu and Swahili are significantly more dispersed than high-resource languages.

**Cross-lingual Similarity:** Since the sentences are perfect translations of each other and the model embeds different languages in a shared region of the embedding space, the $\Gamma$ for all the language pairs should be close to $1/Anisotropy$, however, this is not the case as can be seen in figure 1b for mBERT and A.4 for other models. We observe that the Language Model starts by representing a language in relative isolation, and the more text it sees, the more it contextualizes that language in terms of the other languages, leading to better $\Gamma$ for high-resource languages and languages from same families. The pre-training procedure of models also makes a difference in the language representations, for example, English has high $\Gamma$ on XLMR, possibly because of the cross-lingual pretraining procedure.

**Geometric Separability** It is thought that a language model sharing the same vector space to represent different languages must form an interlingua, however this is not observed. The intra-family $\Phi$ is relatively low for all linguistic families. In the case of mBERT (figure 1c) and XLMR(figure A.4), $\Phi$ of language clusters is extremely high, indicating that the languages form near-isolated vector spaces. In the case of MiniLM(figure A.4), the $\Phi$ for Germanic, Romantic, Slavic and Hellenic families is low, indicating that the word vectors of these languages are assimilated, however, this trend is not observed in the case of other families.

## 3 DISCUSSION AND CONCLUSION

The $\Gamma$ and $\Phi$ values of all the multilingual models clearly show that not all languages are equally represented in the shared embedding space. The low average $\Gamma$ values for low resource languages indicate a serious shortcoming of multilingual models in respect to their cross-lingual capabilities and point out on the heavy dependence of these models on data. The high $\Phi$ values for languages belonging to same language families show that while it is trivial for language subspaces to be separable for individual masked modelling, it is also important for them to have some degree of intersection for better cross-lingual transfer. This observation on $\Phi$ also relate with the results obtained by Philippy et al. (2023) where they show that the distance between language representations in the embedding space of a multilingual model correlate with model's cross-lingual performance. The results we obtain are in line with the work of Conneau et al. (2020b) which prove the correlation of geometric properties of representations with their results when finetuned on downstream tasks.

URM STATEMENT

"The authors acknowledge that all the authors of this work meet the URM criteria of ICLR 2023 Tiny Papers Track."

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

## A  APPENDIX

### A.1  DATASET DETAILS

The XNLI-15 dataset has 112,500 parallel sentences in 15 languages: English (en), French (fr), Spanish (es), German (de), Greek (el), Bulgarian (bg), Russian (ru), Turkish (tr), Arabic (ar), Vietnamese (vi), Thai (th), Chinese (zh), Hindi (hi), Swahili (sw) and Urdu (ur). We sample the first 300 sentences from each dataset and perform our analysis on a vocabulary of 45889 words.

## A.2 MODEL DETAILS

The various transformer based pre-trained models used in our system are as follows:

- **mBERT:** mBERT refers to multilingual BERT trained on 104 languages on the Wikipedia corpus. It is pretrained in a self-supervised fashion, with two pre-training objectives: 1) Masked Language Modelling (MLM) where 15% of the words of a sentence are randomly masked, and the model predicts the masked words. 2) Next Sentence Prediction (NSP): Given a pair of sentences, the model has to predict whether one sentence succeeds the other. We use the 'bert-base-multilingual-cased' implementation of mBERT [1].

- **XLM-RoBERTa:** XLM-RoBERTa is a multilingual version of RoBERTa, pre-trained on 2.5TB of filtered CommonCrawl data in 100 languages. RoBERTa builds on BERT's architecture but uses more data and modifies key hyperparameters such as larger mini-batches and learning rates. It doesn't pretrain on the NSP task. We use the 'xlm-roberta-base' implementation of the XLMR model from HuggingFace. [2].

- **MiniLM:** MiniLM generalises distillation of deep self-attention by using by using self-attention relation distillation for task-agnostic compression of pre-trained Transformers. This eliminates the restriction on the number of attention heads in the student model. The model we use is distilled from XLM RoBERTa base. MiniLM has a significant speed-up compared to it's teacher models and gives a competitive performance. We use the 'Multilingual-MiniLM-L12-H384' implementation of the MiniLM model as found on HuggingFace [3]

## A.3 LINGUISTIC FAMILIES

Table 1: Languages and their Linguistic Families

| Language | Family | Code |
|---|---|---|
| English | Germanic | en |
| German | Germanic | de |
| Hindi | Hindustani | hi |
| Urdu | Hindustani | ur |
| Arabic | Arabic | ar |
| Spanish | Romance | es |
| French | Romance | fr |
| Russian | Slavic | ru |
| Bulgarian | Slavic | bg |
| Swahili | Niger-Congo | sw |
| Thai | Tai | th |
| Vietnamese | Vietic | vi |
| Chinese | Chinese | zh |
| Greek | Hellenic | el |
| Turkic | Turkic | tr |

## A.4 METRICS

**Anisotropy:** Anisotropy of a model is defined as the cosine similarity of any two word embeddings selected on random. For a set of languages $\mathcal{L}$, with $n$ sentences in each language and a model $\mathcal{M}_l$, the anisotropy can be found by taking the average of the cosine similarity of all possible sentence pairs from $|\mathcal{L}| * n$ sentences. The absolute value is taken.

$$Anisotropy(\mathcal{M}_l) = \mod \frac{\sum_{l_u, l_v \in \mathcal{L}} \sum_{i=1}^{n} \sum_{j=i+1}^{n} cosine(s_{l_u,i}, s_{l_v,j})}{\binom{|\mathcal{L}|*n}{2}} \quad (2)$$

---

[1] https://huggingface.co/bert-base-multilingual-cased
[2] https://huggingface.co/xlm-roberta-base
[3] https://huggingface.co/microsoft/Multilingual-MiniLM-L12-H384

**Geometric Separability Index Φ:** calculates the average number of instances that share the same class label as their nearest neighbours.

$$GSI(f) = \frac{\sum_{i=1}^{n} \left( f\left(x_i\right) + f\left(x_i'\right) + 1 \right) \bmod 2}{n} \tag{3}$$

Here, $f$ is a target function that maps instances to their cluster label, $x_i'$ is the nearest neighbour of $x_i$ and $n$ is the total number of data points.

### A.5 ADDITIONAL INFORMATION

**Γ for XLMR and MiniLM**

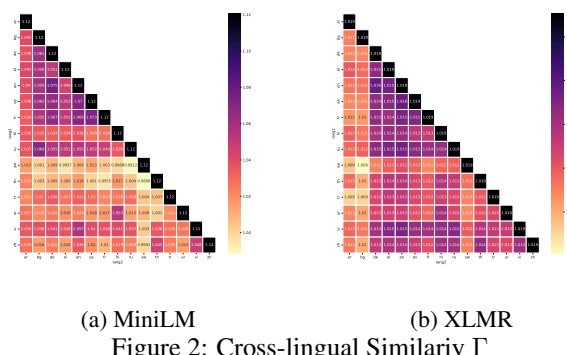

(a) MiniLM                    (b) XLMR

Figure 2: Cross-lingual Similariy Γ

**Φ for XLMR and MiniLM**

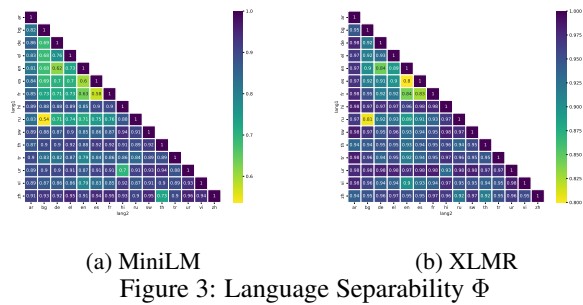

(a) MiniLM                    (b) XLMR

Figure 3: Language Separability Φ

**Other PCA plots**

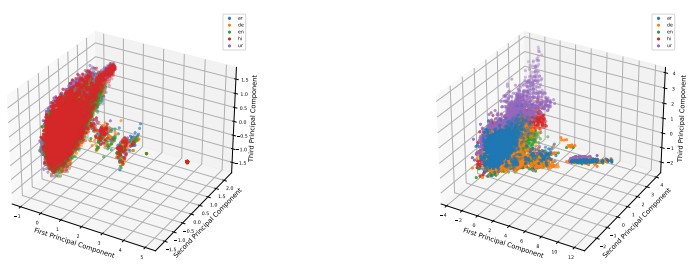

(a) MiniLM                    (b) XLMR

Figure 4: PCA Plots of group (Hindi,Urdu,English,German,Arabic)

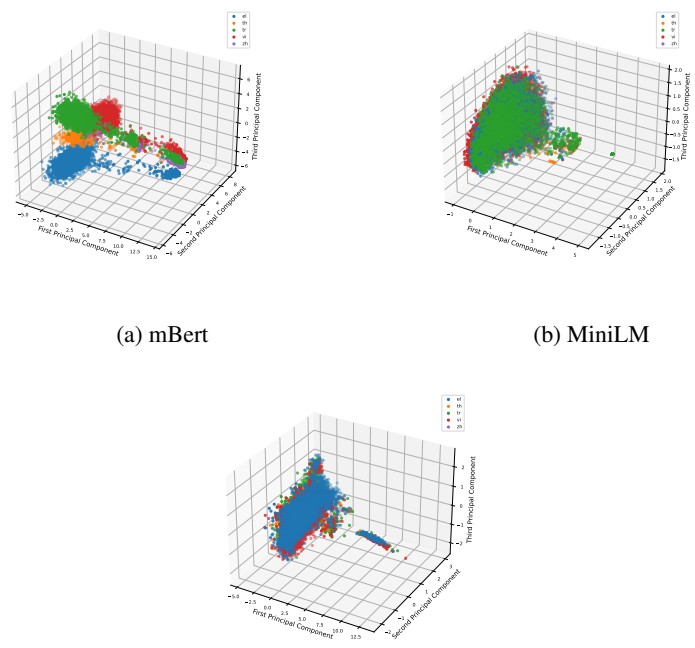

(a) mBert

(b) MiniLM

(c) XLMR

Figure 5: PCA Plots of group (Greek,Turkic,Thai,Vietnamese,Chinese)

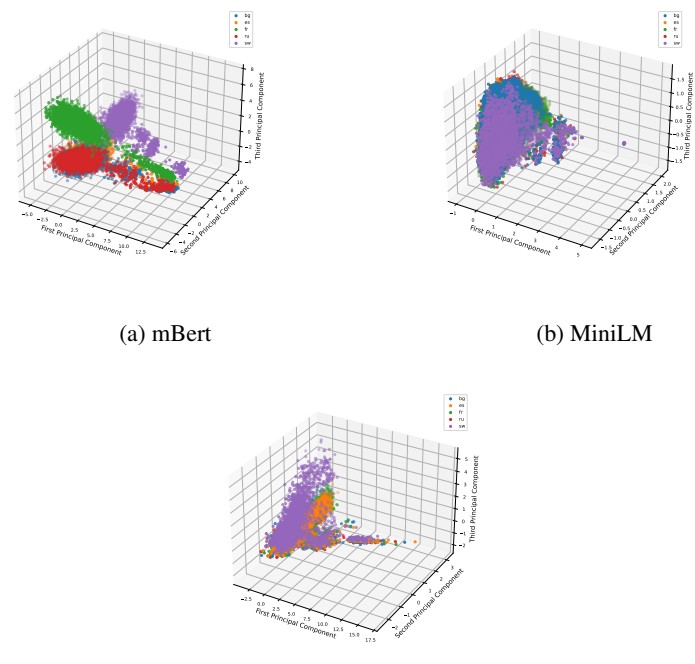

(a) mBert

(b) MiniLM

(c) XLMR

Figure 6: PCA Plots of group (Bulgarian,Russian,Spanish,French,Swahili)

**Cross-lingual Similarity Index ($\Gamma$) values:** These are the exact $\Gamma$ values

Table 2: Γ values of mBert

| lang | ar | bg | de | el | en | es | fr | hi | ru | sw | th | tr | ur | vi | zh |
|------|------|------|------|------|------|------|------|------|------|------|------|------|------|------|------|
| ar | 2.455 | 1.377 | 1.295 | 1.427 | 1.272 | 1.375 | 1.278 | 1.374 | 1.389 | 1.119 | 1.325 | 1.277 | 1.384 | 1.251 | 0.552 |
| bg | 1.377 | 2.455 | 1.483 | 1.519 | 1.409 | 1.485 | 1.43 | 1.361 | 1.652 | 1.121 | 1.262 | 1.358 | 1.232 | 1.389 | 0.523 |
| de | 1.295 | 1.483 | 2.455 | 1.439 | 1.595 | 1.541 | 1.498 | 1.297 | 1.527 | 1.106 | 1.144 | 1.345 | 1.211 | 1.398 | 0.584 |
| el | 1.427 | 1.519 | 1.439 | 2.455 | 1.351 | 1.498 | 1.425 | 1.379 | 1.475 | 1.196 | 1.323 | 1.364 | 1.273 | 1.456 | 0.539 |
| en | 1.272 | 1.409 | 1.595 | 1.351 | 2.455 | 1.59 | 1.574 | 1.236 | 1.495 | 1.048 | 1.096 | 1.283 | 1.173 | 1.473 | 0.606 |
| es | 1.375 | 1.485 | 1.541 | 1.498 | 1.59 | 2.455 | 1.592 | 1.272 | 1.564 | 1.147 | 1.197 | 1.376 | 1.188 | 1.451 | 0.612 |
| fr | 1.278 | 1.43 | 1.498 | 1.425 | 1.574 | 1.592 | 2.455 | 1.266 | 1.487 | 1.019 | 1.197 | 1.301 | 1.154 | 1.423 | 0.563 |
| hi | 1.374 | 1.361 | 1.297 | 1.379 | 1.236 | 1.272 | 1.266 | 2.455 | 1.354 | 1.122 | 1.263 | 1.311 | 1.448 | 1.302 | 0.418 |
| ru | 1.389 | 1.652 | 1.527 | 1.475 | 1.495 | 1.564 | 1.487 | 1.354 | 2.455 | 1.045 | 1.226 | 1.3 | 1.212 | 1.447 | 0.552 |
| sw | 1.119 | 1.121 | 1.106 | 1.196 | 1.048 | 1.147 | 1.019 | 1.122 | 1.045 | 2.455 | 1.167 | 1.206 | 1.235 | 1.143 | 0.367 |
| th | 1.325 | 1.262 | 1.144 | 1.323 | 1.096 | 1.197 | 1.197 | 1.263 | 1.226 | 1.167 | 2.455 | 1.214 | 1.214 | 1.219 | 0.364 |
| tr | 1.277 | 1.358 | 1.345 | 1.364 | 1.283 | 1.376 | 1.301 | 1.311 | 1.3 | 1.206 | 1.214 | 2.455 | 1.254 | 1.343 | 0.563 |
| ur | 1.384 | 1.232 | 1.211 | 1.273 | 1.173 | 1.188 | 1.154 | 1.448 | 1.212 | 1.235 | 1.214 | 1.254 | 2.455 | 1.159 | 0.392 |
| vi | 1.251 | 1.389 | 1.398 | 1.456 | 1.473 | 1.451 | 1.423 | 1.302 | 1.447 | 1.143 | 1.219 | 1.343 | 1.159 | 2.455 | 0.593 |
| zh | 0.552 | 0.523 | 0.584 | 0.539 | 0.606 | 0.612 | 0.563 | 0.418 | 0.552 | 0.367 | 0.364 | 0.563 | 0.392 | 0.593 | 2.455 |

Table 3: Γ values of MiniLM

| lang | ar | bg | de | el | en | es | fr | hi | ru | sw | th | tr | ur | vi | zh |
|------|------|------|------|------|------|------|------|------|------|------|------|------|------|------|------|
| ar | 1.12 | 1.041 | 1.038 | 1.042 | 1.04 | 1.037 | 1.033 | 1.036 | 1.036 | 1.014 | 1.023 | 1.022 | 1.027 | 1.036 | 1.028 |
| bg | 1.041 | 1.12 | 1.061 | 1.057 | 1.058 | 1.061 | 1.055 | 1.035 | 1.068 | 1.0 | 1.002 | 1.03 | 1.021 | 1.037 | 1.016 |
| de | 1.038 | 1.061 | 1.12 | 1.05 | 1.075 | 1.064 | 1.067 | 1.033 | 1.055 | 1.009 | 1.008 | 1.035 | 1.025 | 1.04 | 1.025 |
| el | 1.042 | 1.057 | 1.05 | 1.12 | 1.045 | 1.052 | 1.052 | 1.034 | 1.051 | 0.995 | 1.008 | 1.027 | 1.018 | 1.037 | 1.015 |
| en | 1.04 | 1.058 | 1.075 | 1.045 | 1.12 | 1.07 | 1.069 | 1.038 | 1.054 | 1.009 | 1.016 | 1.033 | 1.023 | 1.057 | 1.036 |
| es | 1.037 | 1.061 | 1.064 | 1.052 | 1.07 | 1.12 | 1.072 | 1.028 | 1.051 | 1.012 | 1.0 | 1.03 | 1.018 | 1.04 | 1.019 |
| fr | 1.033 | 1.055 | 1.067 | 1.052 | 1.069 | 1.072 | 1.12 | 1.028 | 1.048 | 1.002 | 0.995 | 1.022 | 1.016 | 1.038 | 1.009 |
| hi | 1.036 | 1.035 | 1.033 | 1.034 | 1.038 | 1.028 | 1.028 | 1.12 | 1.035 | 0.998 | 1.017 | 1.026 | 1.053 | 1.041 | 1.029 |
| ru | 1.036 | 1.068 | 1.055 | 1.051 | 1.054 | 1.051 | 1.048 | 1.035 | 1.12 | 0.991 | 1.004 | 1.029 | 1.014 | 1.037 | 1.023 |
| sw | 1.014 | 1.0 | 1.009 | 0.995 | 1.009 | 1.012 | 1.002 | 0.998 | 0.991 | 1.12 | 0.988 | 1.004 | 1.009 | 1.002 | 0.999 |
| th | 1.023 | 1.002 | 1.008 | 1.008 | 1.016 | 1.0 | 0.995 | 1.017 | 1.004 | 0.988 | 1.12 | 1.006 | 1.001 | 1.035 | 1.045 |
| tr | 1.022 | 1.03 | 1.035 | 1.027 | 1.033 | 1.03 | 1.022 | 1.026 | 1.029 | 1.004 | 1.006 | 1.12 | 1.02 | 1.028 | 1.024 |
| ur | 1.027 | 1.021 | 1.025 | 1.018 | 1.023 | 1.018 | 1.016 | 1.053 | 1.014 | 1.009 | 1.001 | 1.02 | 1.12 | 1.026 | 1.015 |
| vi | 1.036 | 1.037 | 1.04 | 1.037 | 1.057 | 1.04 | 1.038 | 1.041 | 1.037 | 1.002 | 1.035 | 1.028 | 1.026 | 1.12 | 1.047 |
| zh | 1.028 | 1.016 | 1.025 | 1.015 | 1.036 | 1.019 | 1.009 | 1.029 | 1.023 | 0.999 | 1.045 | 1.024 | 1.015 | 1.047 | 1.12 |

Table 4: Γ values of XLMR

| lang | ar | bg | de | el | en | es | fr | hi | ru | sw | th | tr | ur | vi | zh |
|------|------|------|------|------|------|------|------|------|------|------|------|------|------|------|------|
| ar | 1.019 | 1.011 | 1.01 | 1.012 | 1.011 | 1.01 | 1.01 | 1.011 | 1.011 | 1.009 | 1.011 | 1.009 | 1.01 | 1.011 | 1.011 |
| bg | 1.011 | 1.019 | 1.01 | 1.011 | 1.011 | 1.01 | 1.01 | 1.011 | 1.011 | 1.008 | 1.009 | 1.009 | 1.01 | 1.011 | 1.01 |
| de | 1.01 | 1.01 | 1.019 | 1.014 | 1.015 | 1.014 | 1.014 | 1.013 | 1.014 | 1.011 | 1.012 | 1.012 | 1.012 | 1.013 | 1.013 |
| el | 1.012 | 1.011 | 1.014 | 1.019 | 1.015 | 1.014 | 1.014 | 1.014 | 1.015 | 1.011 | 1.013 | 1.013 | 1.012 | 1.014 | 1.013 |
| en | 1.011 | 1.011 | 1.015 | 1.015 | 1.019 | 1.015 | 1.015 | 1.014 | 1.015 | 1.012 | 1.013 | 1.013 | 1.012 | 1.015 | 1.013 |
| es | 1.01 | 1.01 | 1.014 | 1.014 | 1.015 | 1.019 | 1.014 | 1.013 | 1.014 | 1.011 | 1.012 | 1.012 | 1.012 | 1.014 | 1.013 |
| fr | 1.01 | 1.01 | 1.014 | 1.014 | 1.015 | 1.014 | 1.019 | 1.013 | 1.014 | 1.01 | 1.012 | 1.012 | 1.011 | 1.014 | 1.012 |
| hi | 1.011 | 1.011 | 1.013 | 1.014 | 1.014 | 1.013 | 1.013 | 1.019 | 1.014 | 1.011 | 1.013 | 1.012 | 1.014 | 1.014 | 1.014 |
| ru | 1.011 | 1.011 | 1.014 | 1.015 | 1.015 | 1.014 | 1.014 | 1.014 | 1.019 | 1.011 | 1.013 | 1.012 | 1.012 | 1.014 | 1.013 |
| sw | 1.009 | 1.008 | 1.011 | 1.011 | 1.012 | 1.011 | 1.01 | 1.011 | 1.011 | 1.019 | 1.011 | 1.01 | 1.01 | 1.011 | 1.011 |
| th | 1.011 | 1.009 | 1.012 | 1.013 | 1.013 | 1.012 | 1.012 | 1.013 | 1.013 | 1.011 | 1.019 | 1.012 | 1.012 | 1.013 | 1.014 |
| tr | 1.009 | 1.009 | 1.012 | 1.013 | 1.013 | 1.012 | 1.012 | 1.012 | 1.012 | 1.01 | 1.012 | 1.019 | 1.012 | 1.013 | 1.012 |
| ur | 1.01 | 1.01 | 1.012 | 1.012 | 1.012 | 1.012 | 1.011 | 1.014 | 1.012 | 1.01 | 1.012 | 1.012 | 1.019 | 1.013 | 1.012 |
| vi | 1.011 | 1.011 | 1.013 | 1.014 | 1.015 | 1.014 | 1.014 | 1.014 | 1.014 | 1.011 | 1.013 | 1.013 | 1.013 | 1.019 | 1.014 |
| zh | 1.011 | 1.01 | 1.013 | 1.013 | 1.013 | 1.013 | 1.012 | 1.014 | 1.013 | 1.011 | 1.014 | 1.012 | 1.012 | 1.014 | 1.019 |

**Language Separability ($\Phi$) Values:** These are the exact $\Phi$ values

Table 5: Φ values of mBert

| lang | ar | bg | de | el | en | es | fr | hi | ru | sw | th | tr | ur | vi | zh |
|---|---|---|---|---|---|---|---|---|---|---|---|---|---|---|---|
| ar | 1.0 | 0.986 | 0.987 | 0.985 | 0.989 | 0.985 | 0.988 | 0.991 | 0.986 | 0.991 | 0.988 | 0.99 | 0.991 | 0.991 | 0.989 |
| bg | 0.986 | 1.0 | 0.96 | 0.963 | 0.967 | 0.963 | 0.966 | 0.989 | 0.876 | 0.983 | 0.983 | 0.975 | 0.994 | 0.977 | 0.983 |
| de | 0.987 | 0.96 | 1.0 | 0.965 | 0.904 | 0.929 | 0.934 | 0.989 | 0.963 | 0.967 | 0.975 | 0.954 | 0.992 | 0.957 | 0.976 |
| el | 0.985 | 0.963 | 0.965 | 1.0 | 0.967 | 0.952 | 0.962 | 0.991 | 0.979 | 0.98 | 0.978 | 0.969 | 0.994 | 0.963 | 0.984 |
| en | 0.989 | 0.967 | 0.904 | 0.967 | 1.0 | 0.9 | 0.905 | 0.989 | 0.969 | 0.968 | 0.974 | 0.949 | 0.992 | 0.945 | 0.978 |
| es | 0.985 | 0.963 | 0.929 | 0.952 | 0.9 | 1.0 | 0.869 | 0.99 | 0.958 | 0.969 | 0.978 | 0.958 | 0.993 | 0.952 | 0.975 |
| fr | 0.988 | 0.966 | 0.934 | 0.962 | 0.905 | 0.869 | 1.0 | 0.99 | 0.967 | 0.973 | 0.978 | 0.964 | 0.993 | 0.953 | 0.976 |
| hi | 0.991 | 0.989 | 0.989 | 0.991 | 0.989 | 0.99 | 0.99 | 1.0 | 0.993 | 0.99 | 0.991 | 0.986 | 0.979 | 0.989 | 0.99 |
| ru | 0.986 | 0.876 | 0.963 | 0.979 | 0.969 | 0.958 | 0.967 | 0.993 | 1.0 | 0.995 | 0.993 | 0.985 | 0.996 | 0.982 | 0.985 |
| sw | 0.991 | 0.983 | 0.967 | 0.98 | 0.968 | 0.969 | 0.973 | 0.99 | 0.995 | 1.0 | 0.974 | 0.964 | 0.992 | 0.971 | 0.985 |
| th | 0.988 | 0.983 | 0.975 | 0.978 | 0.974 | 0.978 | 0.978 | 0.991 | 0.993 | 0.974 | 1.0 | 0.971 | 0.994 | 0.981 | 0.96 |
| tr | 0.99 | 0.975 | 0.954 | 0.969 | 0.949 | 0.958 | 0.964 | 0.986 | 0.985 | 0.964 | 0.971 | 1.0 | 0.99 | 0.962 | 0.966 |
| ur | 0.991 | 0.994 | 0.992 | 0.994 | 0.992 | 0.993 | 0.993 | 0.979 | 0.996 | 0.992 | 0.994 | 0.99 | 1.0 | 0.995 | 0.995 |
| vi | 0.991 | 0.977 | 0.957 | 0.963 | 0.945 | 0.952 | 0.953 | 0.989 | 0.982 | 0.971 | 0.981 | 0.962 | 0.995 | 1.0 | 0.972 |
| zh | 0.989 | 0.983 | 0.976 | 0.984 | 0.978 | 0.975 | 0.976 | 0.99 | 0.985 | 0.985 | 0.96 | 0.966 | 0.995 | 0.972 | 1.0 |

Table 6: Φ values of MiniLM

| lang | ar | bg | de | el | en | es | fr | hi | ru | sw | th | tr | ur | vi | zh |
|---|---|---|---|---|---|---|---|---|---|---|---|---|---|---|---|
| ar | 1.0 | 0.821 | 0.859 | 0.831 | 0.814 | 0.841 | 0.85 | 0.893 | 0.833 | 0.894 | 0.871 | 0.895 | 0.887 | 0.89 | 0.911 |
| bg | 0.821 | 1.0 | 0.685 | 0.684 | 0.681 | 0.687 | 0.726 | 0.876 | 0.536 | 0.883 | 0.898 | 0.833 | 0.895 | 0.868 | 0.927 |
| de | 0.859 | 0.685 | 1.0 | 0.764 | 0.615 | 0.703 | 0.712 | 0.884 | 0.707 | 0.874 | 0.896 | 0.817 | 0.899 | 0.863 | 0.919 |
| el | 0.831 | 0.684 | 0.764 | 1.0 | 0.726 | 0.704 | 0.726 | 0.892 | 0.742 | 0.895 | 0.916 | 0.867 | 0.912 | 0.859 | 0.95 |
| en | 0.814 | 0.681 | 0.615 | 0.726 | 1.0 | 0.602 | 0.634 | 0.847 | 0.71 | 0.847 | 0.882 | 0.787 | 0.872 | 0.787 | 0.912 |
| es | 0.841 | 0.687 | 0.703 | 0.704 | 0.602 | 1.0 | 0.576 | 0.895 | 0.746 | 0.855 | 0.918 | 0.84 | 0.908 | 0.833 | 0.943 |
| fr | 0.85 | 0.726 | 0.712 | 0.726 | 0.634 | 0.576 | 1.0 | 0.899 | 0.764 | 0.871 | 0.921 | 0.858 | 0.913 | 0.853 | 0.945 |
| hi | 0.893 | 0.876 | 0.884 | 0.892 | 0.847 | 0.895 | 0.899 | 1.0 | 0.875 | 0.937 | 0.911 | 0.859 | 0.696 | 0.922 | 0.931 |
| ru | 0.833 | 0.536 | 0.707 | 0.742 | 0.71 | 0.746 | 0.764 | 0.875 | 1.0 | 0.905 | 0.89 | 0.84 | 0.908 | 0.874 | 0.902 |
| sw | 0.894 | 0.883 | 0.874 | 0.895 | 0.847 | 0.855 | 0.871 | 0.937 | 0.905 | 1.0 | 0.922 | 0.894 | 0.932 | 0.905 | 0.946 |
| th | 0.871 | 0.898 | 0.896 | 0.916 | 0.882 | 0.918 | 0.921 | 0.911 | 0.89 | 0.922 | 1.0 | 0.887 | 0.939 | 0.901 | 0.732 |
| tr | 0.895 | 0.833 | 0.817 | 0.867 | 0.787 | 0.84 | 0.858 | 0.859 | 0.84 | 0.894 | 0.887 | 1.0 | 0.882 | 0.891 | 0.896 |
| ur | 0.887 | 0.895 | 0.899 | 0.912 | 0.872 | 0.908 | 0.913 | 0.696 | 0.908 | 0.932 | 0.939 | 0.882 | 1.0 | 0.933 | 0.955 |
| vi | 0.89 | 0.868 | 0.863 | 0.859 | 0.787 | 0.833 | 0.853 | 0.922 | 0.874 | 0.905 | 0.901 | 0.891 | 0.933 | 1.0 | 0.94 |
| zh | 0.911 | 0.927 | 0.919 | 0.95 | 0.912 | 0.943 | 0.945 | 0.931 | 0.902 | 0.946 | 0.732 | 0.896 | 0.955 | 0.94 | 1.0 |

Table 7: Φ values of XLMR

| lang | ar | bg | de | el | en | es | fr | hi | ru | sw | th | tr | ur | vi | zh |
|---|---|---|---|---|---|---|---|---|---|---|---|---|---|---|---|
| ar | 1.0 | 0.946 | 0.98 | 0.966 | 0.971 | 0.971 | 0.978 | 0.984 | 0.967 | 0.985 | 0.935 | 0.984 | 0.978 | 0.98 | 0.939 |
| bg | 0.946 | 1.0 | 0.92 | 0.916 | 0.9 | 0.916 | 0.95 | 0.971 | 0.805 | 0.971 | 0.943 | 0.961 | 0.978 | 0.958 | 0.948 |
| de | 0.98 | 0.92 | 1.0 | 0.932 | 0.845 | 0.911 | 0.916 | 0.973 | 0.919 | 0.948 | 0.949 | 0.936 | 0.981 | 0.939 | 0.958 |
| el | 0.966 | 0.916 | 0.932 | 1.0 | 0.892 | 0.897 | 0.922 | 0.971 | 0.925 | 0.965 | 0.95 | 0.954 | 0.981 | 0.939 | 0.967 |
| en | 0.971 | 0.9 | 0.845 | 0.892 | 1.0 | 0.8 | 0.828 | 0.96 | 0.89 | 0.931 | 0.927 | 0.916 | 0.972 | 0.899 | 0.952 |
| es | 0.971 | 0.916 | 0.911 | 0.897 | 0.8 | 1.0 | 0.828 | 0.976 | 0.914 | 0.94 | 0.944 | 0.935 | 0.981 | 0.926 | 0.952 |
| fr | 0.978 | 0.95 | 0.916 | 0.922 | 0.841 | 0.828 | 1.0 | 0.977 | 0.934 | 0.953 | 0.951 | 0.951 | 0.985 | 0.937 | 0.97 |
| hi | 0.984 | 0.971 | 0.973 | 0.971 | 0.96 | 0.976 | 0.977 | 1.0 | 0.966 | 0.983 | 0.957 | 0.972 | 0.931 | 0.975 | 0.965 |
| ru | 0.967 | 0.805 | 0.919 | 0.925 | 0.89 | 0.914 | 0.934 | 0.966 | 1.0 | 0.975 | 0.935 | 0.96 | 0.98 | 0.95 | 0.919 |
| sw | 0.985 | 0.971 | 0.948 | 0.965 | 0.931 | 0.94 | 0.953 | 0.983 | 0.975 | 1.0 | 0.955 | 0.953 | 0.983 | 0.951 | 0.97 |
| th | 0.935 | 0.943 | 0.949 | 0.95 | 0.927 | 0.944 | 0.951 | 0.957 | 0.935 | 0.955 | 1.0 | 0.946 | 0.969 | 0.939 | 0.916 |
| tr | 0.984 | 0.961 | 0.936 | 0.954 | 0.916 | 0.935 | 0.951 | 0.972 | 0.96 | 0.953 | 0.946 | 1.0 | 0.972 | 0.948 | 0.949 |
| ur | 0.978 | 0.978 | 0.981 | 0.981 | 0.972 | 0.981 | 0.985 | 0.931 | 0.98 | 0.983 | 0.969 | 0.972 | 1.0 | 0.983 | 0.965 |
| vi | 0.98 | 0.958 | 0.939 | 0.939 | 0.899 | 0.926 | 0.937 | 0.975 | 0.95 | 0.951 | 0.939 | 0.948 | 0.983 | 1.0 | 0.955 |
| zh | 0.939 | 0.948 | 0.958 | 0.967 | 0.952 | 0.952 | 0.97 | 0.965 | 0.919 | 0.97 | 0.916 | 0.949 | 0.965 | 0.955 | 1.0 |

