# OpenReview forum: "The Geometry of Multilingual Language Models: An Equality Lens"
_ICLR.cc/2023/TinyPapers — Submitted to Tiny Papers @ ICLR 2023_

### Official Review · Reviewer_QgQ5 · 2023-03-19

**Confidence:** 5

**Summary Of Contributions:**

Analysis of the geometry of three multilingual language models in Euclidean space is presented, with findings that indicate all languages are represented by unique geometries. Cross-Lingual Similarity Index to study the semantic similarity across Languages is also presented.

**Rating:**

Clear, Correct, and Reproducible (CCR): a submission which meets the reviewing criteria

**Strengths And Weaknesses:**

Very interesting topic, with lots of plots. The author did a decent job fitting all the plots in 2 pages, but the findings are unclear, especially in Figures 3b and c. Also not sure where Figures 1 and 2 are or an introduction.

Clarity on findings is needed. Even with the Appendix table, there are a collection of numbers but what do they mean? Maybe A sentence in Appendix to explain this would be great.


**Suggested Changes:**

The author should decide the most important message he/she wants to get across and emphasise what that is. It appears it was a very quick job of converting a full-length paper to a short one. Although the topic is interesting and much welcomed in an initiative such as a “tiny” paper, clarity is required.

This paper can be cleaned up, and I strongly suggest the authors do so. Decide if all three methods are needed. Maybe choose two and use that space to provide a clear vision.

---

### Official Review · Reviewer_rBAb · 2023-03-30

**Confidence:** 4

**Summary Of Contributions:**

The authors analyze the word embedding representations from 3 different multilingual language models. They introduce a new metric, called Cross-Lingual Similarity Index, to quantify the semantic similarity across languages.

**Rating:**

Clear, Correct, and Reproducible (CCR): a submission which meets the reviewing criteria

**Strengths And Weaknesses:**

*Strengths*
- This study tackles an interesting question and compares the geometry of 3 different models.
- The analysis takes into account the anisotropy, an important aspect of embedding spaces.
- The authors introduce a novel metric, the Cross-Lingual Similarity Index.

*Weaknesses*
- The correctness of the paper would be improved if modifying the title or including a few comparisons in the text (see my comment in suggestions). This is because the current state of the paper does not analyse fairness of the models. The "fairness lens" in the title makes the reader to expect a different sort of analysis, and it can also result in the paper not reaching the right audience.
- The clarity of the paper could be improved: I miss references to relevant literature, specially when communicating your findings (how do they relate to other works?). Also, the last sentence in the abstract is not clear (what does it mean that a representation is "significantly low"?).
- In line with the previous point, I think the takeaway message from the paper is not clear. How do you combine the results from the 3 metrics to make a conclusion? The discussion in paragraphs "Cross-lingual Similarity" and "Geometric Separability" goes back and forth between models, language clusters and language families, which makes it hard to follow. The conclusion should remind the reader the main goal of the study and emphasise what are your findings.

**Suggested Changes:**

- I would strongly recommend to modify the title, so it is not misleading: "a fairness lens" is a big claim that is not met in the paper. Otherwise, this could also be address in the text by performing a small comparison between best and worst-off language groups, e.g., looking at min-max gaps.
- Connect the story you tell in "Cross-lingual Similarity" and "Geometric Separability". Also, if the Cross-Lingual Similarity Index measures semantic similary, the following claim does not seem to support that: "Γ for Chinese is significantly low for mBERT which might be because Chinese is a purely ideographic language".
- All figures should be referenced in the text. Currently, Figure 2 and 3 are not referenced.
- I would simplify all the figures as they are hard to interpret, e.g., they are too tiny. The legends are not readable in any figure. For example, you could simplify it all if showing plots for only 1 model and place the others in the Appendix. This way you would also gain space for writing.
- Normalize the writing. e.g, XLM-R or XLMR? Also, no need to capitalize 'Language Model'.
- In equation 1, what does it mean the subindex 'l' in M_l? What is the range of values that Anisotropy(M_l) can take?
- It would be interesting, perhaps as future work, to extend the similarity index to quantify the cross-lingual similarity of a model (not only between language pairs).
- I would maybe discard the future work to gain space to make a more solid conclusion.
- Recommended references:

multilinguality with a fairness lens:

https://www.microsoft.com/en-us/research/uploads/prod/2021/01/5089.ChoudhuryM.pdf

https://aclanthology.org/2022.coling-1.318/

geometry of multilingual language models:

https://aclanthology.org/D19-6106/

https://aclanthology.org/2020.repl4nlp-1.16.pdf

https://aclanthology.org/2022.findings-acl.103.pdf

---

### Author Response · Authors · 2023-05-30
**wish to opt-in for archival**

We wish to opt-in for archival and adhere that all requirements are satisfied

---

### Meta-Review · Area_Chair_bqUD · 2023-04-04

**Recommendation:** Invite to present
**Confidence:** 5

**Metareview:**

The authors analyse the word embedding representations from 3 different multilingual language models. They introduce a new metric, called Cross-Lingual Similarity Index, to quantify the semantic similarity across languages.

The paper does not analyse the fairness of the models. Clarify the findings better. Avoid using phrases such as “significantly low” unless you are willing to provide a statistically significant ranking. The takeaway message from the paper is not clear.

However, with some minor edits, this paper can be improved.


**Summary:**

The authors analyse the word embedding representations from 3 different multilingual language models. They introduce a new metric, called Cross-Lingual Similarity Index, to quantify the semantic similarity across languages.

**Comments And Feedback To The Authors:**

Read the reviews carefully. Try to incorporate the changes.

**Reason For Not Giving A Higher Recommendation:**

N/A

**Reason For Not Giving A Lower Recommendation:**

The paper is CCR. It needs some minor changes. If the authors make these corrections then the paper will be a good addition.

---

> ### Author Response · Authors · 2023-05-30
> **Changes Made as suggested**
>
> -Removed the statistical significant sentence
> -Changed title
> -Modified conclusion to make the theme of the paper clearer

---

### Decision · Program_Chairs · 2023-04-07

Invite to present